# Enterotoxin Genes, Antibiotic Susceptibility, and Biofilm Formation of Low-Temperature-Tolerant *Bacillus cereus* Isolated from Green Leaf Lettuce in the Cold Chain

**DOI:** 10.3390/foods9030249

**Published:** 2020-02-25

**Authors:** Kyung Min Park, Hyun Jung Kim, Mooncheol Jeong, Minseon Koo

**Affiliations:** 1Food Biotechnology, Korea University of Science & Technology, Daejeon 34113, Korea; kyungmuni82@naver.com (K.M.P.); hjkim@kfri.re.kr (H.J.K.); 2Consumer Safety, Korea Food Research Institute, Wanju-gun 55365, Korea; mcjeong@kfri.re.kr; 3Food Analysis Center, Korea Food Research Institute, Wanju-gun 55365, Korea

**Keywords:** low-temperature tolerant, *Bacillus cereus*, enterotoxin gene, antibiotic susceptibility, biofilm formation, cold chain

## Abstract

The prevalence and characteristics of low-temperature-tolerant *Bacillus cereus* (psychrotolerant *B. cereus*) in green leaf lettuce collected during cold chain were investigated. Among the 101 isolated *B. cereus* samples, only 18 were capable of growth at 7 °C, and these isolates shared potential health hazard characteristics with mesophilic isolates. Most psychrotolerant *B. cereus* isolates contained various combinations of *nheA, nheB, nheC,*
*hblA, hblA, hblC, hblD*, *cytK,* and *entFM.* Most isolates of psychrotolerant *B. cereus* possessed at least two enterotoxin genes and 28% of isolates harbored tested nine enterotoxin genes. Additionally, the psychrotolerant *B. cereus* isolates showed resistance to tetracycline and rifampin and intermediate levels of resistance to clindamycin. A total of 23% of isolates among psychrotolerant *B. cereus* displayed a high level of biofilm formation at 7 °C than at 10 °C or 30 °C. The results of this study indicate that cold distribution and storage for green leaf lettuce may fail to maintain food safety due to the presence of enterotoxigenic, antibiotic-resistant, and strong biofilm forming psychrotolerant *B. cereus* isolates, which therefore poses a potential health risk to the consumer. Our findings provide the first account of the prevalence and characteristics of psychrotolerant *B. cereus* isolated from green leaf lettuce during cold storage, suggesting a potential hazard of psychrotolerant *B. cereus* isolates to public health and the food industry.

## 1. Introduction

Temperature is one of the crucial factors affecting the safety and quality of vegetables and fruits [1]. To inhibit the growth of foodborne pathogens, vegetables must be maintained at refrigeration temperatures throughout all steps from harvest to consumption. The temperature of perishable foods such as fruits and vegetables must be controlled throughout the supply chain using cold chain systems. However, some bacteria can grow at refrigeration temperatures, and these bacteria (called phychrotolerant bacteria) have been recognized as spoilage bacteria or foodborne pathogens that cause food safety and food quality problems during refrigerated storage and the distribution process of various food products [2,3].

*Bacillus cereus* is an endospore-forming foodborne pathogen causing diarrheal or emetic food poisoning [4]. *B. cereus* is ubiquitous in soil, rive, spices, milk, and vegetables and on human skin [4]. In a previous study on the microbiological quality of fresh vegetables, the most frequent foodborne pathogen found in fresh vegetables was *B. cereus*, with a contamination rate of 37.5% [5]. The emetic type of food poisoning is caused by strains producing a heat-stable toxin called cereulide, which is encoded by the *ces* gene [6] and is produced in food [7]. The diarrheal type of food poisoning is caused by strains producing one or several enterotoxins produced during the growth of vegetative cells in the small intestine [8]. The enterotoxins produced by *B. cereus* have been described: hemolysin BL (HBL), consisting of three genes of *hblA, hblC,* and *hblD*; nonhemolytic enterotoxin (NHE), encoded by *nheA, nheB,* and *nheC*; cytotoxin K (CytK); and enterotoxin FM (EntFM). Although *B. cereus* is a well-known cause of food poisoning, it is also a prolific producer of lipases and proteases and is recognized as causing spoilage in dairy products and refrigerated products [9]. Furthermore, *B. cereus* can attach to food contact surfaces such as pipelines and stainless steel equipment where it might germinate under optimal growth conditions [10].

The *B. cereus* group was not traditionally considered as a psychrotolerant species until some cold-tolerant isolates were identified in 1990. The term psychrotrotolerant is actually used for mesophlic bacteria that are low temperature tolerant and grow in a psychrophilic temperature range. Optimum growth temperature for psychrophilic bacteria is 15 °C, while psychrotolerant bacteria have a wider range as they can grow at 25–40 °C, similar to their mesophilic counterparts, and can grow at temperatures at 7 °C and below [11,12,13]. At low temperatures, psychrotolerant *B. cereus* has slower metabolic rates but higher catalytic efficiencies than mesophilic strains [11]. Psychrotolerant *B. cereus* is a contaminant in many raw materials used for food production and cold-stored foods [12,13] and can reach levels that are potentially harmful for human health during refrigerated storage [14]. Some psychrotolerant *B. cereus* strains isolated from whole liquid egg product were able to produce toxins in the food at 6, 8, and 10 °C [15] and were found to contain the *ces* gene responsible for cereulide production. Psychrotolerant *B. cereus* are still virulent as they express toxin genes and induce cytotoxic activity to the same extent as *B. cereus* mesophilic strains at low temperatures [16]. Therefore, psychrotolerant *B. cereus* should be identified as a potential hazard for refrigerated foods that are subjected to temperature abuse that occurs during food shipping, distribution, and storage. These bacteria also show high resistance to other environmental stresses. In a recent study on psychrotolerant *B. cereus*, after refrigerated storage at 8 °C and a pH of 3–4, food products contained a high number of vegetative psychrotolerant *B. cereus* cells; the *B. cereus* strains were able to grow to an approximately 8 log colony forming unit (CFU)/g after seven days [17]. Another strain of psychrotolerant *B. cereus* was able to grow at 10 °C under anaerobic conditions and reached 4 to 6 log CFU/g [18]. Due to their potential to grow at low temperature, their ability to produce toxins, cross protection phenomenon to other environmental stresses, and their implications in foodborne outbreaks, psychrotolerant isolates of *B. cereus* have been of concern in the food industry.

*B. cereus* is sometimes connected to hospital infection such as bacteremia, central nervous system infection, respiratory infections, endocarditis, and food poisoning. The occurrence of hospital infection by *B. cereus* is low, but the mortality is high, regardless of aggressive treatment with antibiotics. Thus, the spread of antimicrobial resistant bacteria is a severe public health problem and is associated with increasing mortality and medical costs [19,20]. Most *B. cereus* isolates are susceptible to commonly used antibiotic agents except on beta-lactam antibiotics [21], however, prolonged and extensive antibiotic use has led to the emergence of single or multidrug resistant bacteria. Some *B. cereus* isolated from foodstuff have been reported to be resistant to tetracycline, erythromycin, and rifampin [22]. However, little is known regarding the antimicrobial resistance of psychrotolerant *B. cereus* isolated from raw vegetables. Therefore, the antimicrobial resistance of *B. cereus* isolated from foodstuff must be monitored for human health and food safety.

The presence of *B. cereus* strains in green leafy vegetables were frequently observed [4,21,23]. Considering green leafy vegetables are generally distributed throughout the cold chain, there is the possibility of the existence of psychrotolerant *B. cereus* strains in these types of food. Furthermore, since vegetables are consumed without further cooking, contamination of psychrotolerant *B. cereus* can remain viable, but little study was available. Only a few studies have provided information about the prevalence and characteristics of psychrotolerant *B. cereus* strains from foodstuff such as milk, dairy products, and chilled food [17,18,24]. Therefore, the goal of the present study was to evaluate the presence and toxigenic characteristics of *B. cereus* strains, especially psychrotolerant *B. cereus* strains in green leaf lettuce as an example of leafy vegetables distributed in cold chain. We investigated the presence of genes encoding enterotoxins, antibiotic susceptibility, and biofilm formation ability at refrigeration temperatures. The results from this study will provide an enhanced understanding of the behavior of psychrotolerant *B. cereus* strains in agricultural products distributed through the cold chain.

## 2. Materials and Methods

### 2.1. Collection of Samples

We collected the green leaf lettuce from every step of the “farm-to-retail” distributions of leafy vegetables. More detailed sampling steps were as follows: Green leaf lettuce was harvested in an open field on a farm located in Gyeongii Province, South Korea. Harvested vegetables were transported to a packing house located within the same farm where they were hand-packed into boxes at a low temperature. Samples for microbiological quality analysis were collected after harvesting (referred to as ‘harvest’) and packaging (‘packaging’). On the same day, the packaged boxes were loaded onto refrigerated trucks for transportation and storage in a distribution center. The next samples were collected after arrival at the distribution center (‘transportation’) and indicated storage duration at 9–10 °C in the distribution center, prior to shipment to the retail outlet (‘distribution center’). The final sampling site (‘retail shop’) was the point of sale in the retail shop.

The vegetable samples (100 g) from each sampling site were individually placed in sterilized bags from ten different boxes of the same lot code and were maintained at 5 °C during transport to the laboratory for microbiological quality analysis. The temperature of the green leaf lettuce was monitored with a continuous temperature recorder connected to a reader (BTM-4208SD, LT Lutron, Taipei, Taiwan) with a thermocouple. The thermocouple was completely inserted into the center of the vegetables. The ambient temperature ranged from 0 °C to 5 °C and the temperature of the vegetables was maintained at 5 °C ± 2 °C throughout the entire distribution chain (Figure 1).

### 2.2. Microbiological Analysis for the B. cereus Group

The microbiological procedures for *B. cereus* group isolation recommended by the Korean food code [25] were followed. To isolate the *B. cereus* group from green leaf lettuce collected during the supply chain, 25 g of each sample was transferred to a sterile stomacher bag with 225 mL of 0.85% saline and homogenized for 2 min at 230 rpm in a stomacher 400 (Seward, Norfolk, UK). Next, the suspension solution was inoculated onto mannitol–egg-yolk–polymyxin B agar (MYP, Merck, Darmstadt, Germany) and incubated at 30 °C for 18 h. The *B. cereus* group appears as dry, pinkish colonies surrounded by a ring of dense precipitate due to the degradation of lecithin. Suspected colonies were selected, and a maximum of five typical colonies from each sample were subcultured on tryptic soy agar (TSA, Merck Darmstadt, Germany). Where there were less than five colonies, all were isolated. Biochemical identification of the selected colonies was conducted by using the Vitek-II system with the BCL card (bioM´erieux, Inc., Marcy l’Etoile, France), according to the manufacturer’s directions.

### 2.3. Bacterial DNA Extraction

To obtain the genomic DNA, 1 mL of the 18–20 h bacterial culture was centrifuged (15,000× *g*, 5 min) and the pellet was washed twice with 1 mL phosphate-buffered saline (PBS). The cell pellet was resuspended in 100 µL sterilized distilled water and heated at 95 °C for 10 min and was then placed on ice. After centrifugation at 14,000× *g* for 2 min, the supernatant was used as the template for molecular identification and the toxin gene profile. 

### 2.4. Identification of B. cereus and B. thuringiensis from the B. cereus Group

To isolate *B. cereus* and *B. thuringiensis*, Polymerase chain reaction (PCR) amplification of the gyrase B (*gyrB*) gene and crystal (*cry*) gene was carried out as described by Yamada et al. [26]. The primers used for the identification of *B. cereus* were BC1 (5′-ATTGGTGACACCGATCAAACA-3′) and BC2r (5′-TCATACGTATGGATGTTATTC-3′). BTJH-1F (GCTTACCAGGGAAATTGGCAG) and BTJH-R (ATCAACGTCGGCGTCGG) were used for the identification of *B. thuringiensis* from the *B. cereus* group. The specific primers for the *cry* gene detection of *B. thuringiensis* were K3 (5’-GGCTGTGACACGAAGGATATAGCCAC-3’) and K5 (5’-AGGACCAGGATTTACAGGAGG-3’). The amplification of *gyrB* (365 bp) for *B. cereus*-specific fragments and *gyrB* (299 bp) and *cry* for *B. thuringiensis* (1600 to 1700 bp) was performed using 30 cycles, each consisting of 60 s at 94 °C, 90 s at 58 °C (*gyrB* for *B. cereus* and *cry* gene for *B. thurigiensis*), 90 s at 63 °C (*gyrB* for *B. thurigiensis*), and 90 s at 72 °C, with a final extension step at 72 °C for 7 min. *Bacillus mycoides* were identified based on the distinct characteristics of filamentous and rhizoid colonies that formed on the agar plate [27].

### 2.5. Growth Properties

To determine the psychrotolerant properties, all isolates of the *B. cereus* group were screened for their growth ability at 10 °C, 7 °C, and 5 °C. The *B. cereus* group isolates were first inoculated on tryptic soy agar and incubated at 30 °C for 18 h. One colony of overnight culture was inoculated on a fresh TSA plate and incubated at 10 °C for 12 days, at 7 °C for 20 days, and at 5 °C for 20 days. After incubation, the plates were examined for observable growth.

### 2.6. Detection of Enterotoxin and Emetic Toxin Genes

To assess the putative pathogenic potential of the *B. cereus* group isolates, all isolates were screened by PCR for the presence/absence of toxin genes known to encode enterotoxins and emetic toxins. The primer pairs for each toxin gene used in this study were based on our previous study [18]. The primer pair sequences and the conditions of amplification are listed in Table 1. *B. cereus* ATCC 14579 (diarrheagenic) and NCCP 14796 (emetic) strains were used as the control strains.

### 2.7. Antibiotic Susceptibility Testing

All isolates were tested using the standard disk diffusion method on Mueller–Hinton agar (Merck), as described by the Clinical and Laboratory Standards Institute [23]. The antimicrobial agents were tested and their concentrations were as follows: cefotaxime (30 µg), ceftriaxone (30 µg), chloramphenicol (30 µg), ciprofloxacin (5 µg), clindamycin (2 µg), erythromycin (15 µg), gentamicin (10 µg), imipenem (10 µg), penicillin (10 µg), rifampin (5 µg), tetracycline (30 µg), and vancomycin (10 µg). After incubation at 30 °C for 24 h, the antibiotic susceptibility was measured, and the results were interpreted as “susceptible”, “intermediate”, and “resistant” in accordance with the criteria provided by the Clinical and Laboratory Standards Institute [28]. Quality control was performed using the reference strain *Staphylococcus aureus* ATCC 29213.

### 2.8. Quantification of Biofilm Formation

The biofilm formation ability of the *B. cereus* group isolates was performed according to the modified method by Singh et al. [29] and was investigated under different temperature conditions (30 °C, 10 °C, and 7 °C) using the microtiter plate method. These temperatures were chosen due to their importance in the distribution chain of agricultural products: 30 °C is the optimum temperature for growth of the *B. cereus* group, 10 °C is the average surface temperature of the vegetables in the distribution chain, and 7 °C is the growth temperature of the psychrotolerant *B. cereus* group isolates. To evaluate biofilm formation, all *B. cereus* group isolates were inoculated into a test tube containing 10 mL of tryptic soy broth (TSB, Merck) and were incubated for 18 h at 30 °C. A total of 100 µL of each bacterial suspension was inoculated into 900 µL of fresh tryptic soy broth in three wells of three sterile 48-well flat-bottomed polystyrene plates. Negative control wells containing only TSB were inoculated in triplicate.

For biofilm evaluation, we used biological triplicates. The plates were covered and incubated aerobically for 72 h at 30 °C, for 12 days at 10 °C, and for 20 days at 7 °C. After incubation, the contents of the microtiter plate were poured off, and the wells were washed three times with 1 mL of phosphate buffered saline. The attached bacteria were then fixed with 1 mL of methanol per well for 15 min, after which the plates were emptied and dried at room temperature. Then, the plates were stained with 500 µL of 1% crystal violet per well for 10 min. The stain was removed, and the plates were washed under running tap water. The plates were air dried. In this study, sterile broth and *B. cereus* 14579 served as the negative control and *B. cereus* ATCC 10876 was used as the positive control strain for all incubation temperatures.

The optical density (OD) of each well was measured at 550 nm with a microplate reader (Synergy™ Mx, BioTek, Winooski, VT, USA). The OD of each strain was obtained from the mean value of the three respective wells. The strains were classified as follows. The cut-off OD (ODc) for the microtiter plate test was defined as three standard deviations above the mean OD of the negative control. The strains were classified into four categories: no biofilm producer (ODs ≤ ODc), weak biofilm producer (ODc < ODs ≤ 2 × ODc), moderate biofilm producer (2 × ODc < ODs ≤ 4 × ODc), or strong biofilm producer (4 × ODc < ODs).

### 2.9. Statistical Analysis

All assays were performed in triplicate in two independent experiments, and the results were expressed as the average and log-transformed values. SPSS statistical software (IBM, Armonk, NY, USA) was used to evaluate the results by one-way analysis of variance (ANOVA). To compare the means, Duncan’s test was used with a significance level of *p* < 0.05.

## 3. Results and Discussion

### 3.1. Prevalence and Identification of the B. cereus Group

The prevalence and identification of the *B. cereus* group from green leaf lettuce throughout the cold chain are illustrated in Table 2. The *B. cereus* group contaminated all samples collected at each sampling point, and the average contamination level was less than 3 log CFU/g. After harvest, the mean *B. cereus* number was 1.9 log CFU/g, and the level (2.9 log CFU/g) of contamination at the packaging point was 1 log CFU/g higher than that at the harvest point. During cold storage in the distribution center, the contamination level of *B. cereus* was maintained, but the average count at the transportation point decreased to 2.1 log CFU/g from 2.6 log CFU/g at the distribution center. Few studies have evaluated *B. cereus* contamination in vegetables, and approximately 50% of the tested vegetable products were contaminated with *B. cereus* [21,30,31,32]. *B. cereus* contamination of vegetables may be common, and the consumption of vegetables contaminated with *B. cereus* may result in exposure to foodborne illness [32].

In fresh-cut vegetable end products, *B. cereus* levels between 4 and 5 log CFU/g are considered unsatisfactory for consumption, according to the Health Protection Agency guidelines of the United Kingdom for ready-to-eat foods [33]. All samples collected from each point during the cold chain had values less than 3 log CFU/g, but there is a potential possibility of bacterial contamination in raw fresh vegetables and the risk of foodborne illness outbreak by pathogens is greater when compared to cooked foods as raw vegetables are most often consumed without cooking. Conventional PCR was performed with 101 *B. cereus* group isolates, and the results supported the identification of *B. cereus* as all isolates were non-rhizoidal cells and did not harbor the *cry* gene or the specific *gyrB* gene for *B. thuringiensis*.

Low temperatures in the cold chain can effectively control the proliferation of foodborne pathogens [34]. However, the use of long-term cold storage may induce changes in the cellular processes, cell membranes, cell surface protein expression, and ribosome structures that promote efficient survival and proliferation under food environment-associated stress conditions [35]. Psychrotolerant *B. cereus* isolates, in comparison to non-stress-adapted bacteria, can more easily adapt to other stresses such as pH, heat, and anaerobic conditions [17,18]. Thus, monitoring the prevalence of psychrotolerant *B. cereus* isolates in vegetables throughout the cold chain could be an important food safety factor for predicting the occurrence of outbreaks caused by psychrotolerant *B. cereus* in fresh vegetables.

### 3.2. Psychrotolerant Properties of B. cereus Isolated from Vegetables

We collected a total of 101 *B. cereus* isolates from green leaf lettuce samples. All isolates did not grow at 5 °C: 58.4% among the 101 isolates grew at 12 °C, 50.5% at 10 °C, 17.8% at 7 °C, and none at 5 °C (Table 3). Among the 18 isolates that grew at 7 °C, the majority (17 isolates, 98%) of isolates were collected from green leaf lettuce during transportation from a distribution center to a retail shop. More than 50% of the isolates could grow at 12 °C or 10 °C, and approximately 18% of the isolates were capable of growth at 7 °C; therefore, psychrotolerant *B. cereus* isolates from green leaf lettuce could increase, depending on the storage condition and transport period of the products under low temperatures, considering that refrigeration is the main conservation technology of these products. The psychrotolerant *B. cereus* isolates may be important in the food industry due to the fact that these isolates are related to the reduction of the shelf life of food products [36], or the increase in the number of microorganisms, which in turn can become a food poisoning problem.

The psychrotolerant *B. cereus* group is detected in soil, milk, ice cream, dairy products, rice salad, and spices [37,38]. Psychrotrophic *B. cereus* isolates can become a food poisoning problem when these isolates possess toxin genes to produce diarrheic or emetic syndromes. Recent studies have shown that psychrotolerant *B. cereus* isolates carry virulence toxin genes associated with diarrheal disease (*nheACD* and *hblACD*), and psychrotolerant *B. cereus* isolates have been shown to produce both HBL and NHE toxins [39,40,41]. Further studies are necessary to determine the prevalence of psychrotolerant *B. cereus* isolates in various agricultural products supplied throughout the cold chain to evaluate virulence factors, antibiotic resistance, and biofilm formation of these isolates as well as to predict the possible occurrence of foodborne illness.

### 3.3. Enterotoxigenic Potential of Psychrotolerant B. cereus Isolates

HBL, NHE, CytK, EntFM, and cereulide (emetic toxin) are major virulence factors produced by *B. cereus* [42]. To evaluate the food poisoning potential of psychrotolerant *B. cereus* isolates, we investigated the distribution and toxigenic profile of toxin genes, and the results are shown in Table 4. NHE is produced only when all three NHE enterotoxin (*nheA, nheB,* and *nheC*) complexes are present [43]. We found that 98% of the isolates of non-psychrotolerant *B. cereus* and 94% of the psychrotolerant *B. cereus* isolates harbored the NHE complex (the combination of *nheA, nheB,* and *nheC*). The production of the HBL enterotoxin composed of B, L2, and L1 subunits depends on the expression of all three genes, namely, *hblC*, *hblD,* and *hblA*. Thirty percent of the non-psychrotolerant *B. cereus* isolates and 44% of the psychrotolerant *B. cereus* isolates possessed three coding genes of the HBL complex. The *cytK* gene, which might cause necrotic enteritis, was detected in 96% of the mesophilic and 57% of the psychrotolerant *B. cereus* isolates. In addition, 88% of the non-psychrotolerant *B. cereus* isolates and 78% of the psychrotolerant *B. cereus* isolates possessed the *entFM* gene. None of the isolates possessed the *ces* gene. A total of 23% of the non-psychrotolerant *B. cereus* isolates and 28% of the psychrotolerant *B. cereus* isolates possessed all tested enterotoxin genes.

We observed that 18 isolates capable of growth at 7 °C among the 101 *B. cereus* isolates may have the potential to proliferate in food in response to changes in the optimal environmental conditions of these bacteria or in long-term cold storage. Bartoszewicz et al. [37] reported that psychrotolerant *B. cereus* harbors well-known toxigenic-associated genes such as *nheA, hblA,* and *cytK*. Most psychrotolerant *B. cereus* isolates have been reported to possess the enterotoxin gene and show high hemolytic activity [40,44]. Our data are consistent with those of previous reports showing that psychrotolerant *B. cereus* can grow at refrigeration temperatures and possesses at least two enterotoxin genes. These results suggest that psychrotolerant *B. cereus* isolates from green leaf lettuce may pose a health risk if the enterotoxin genes are expressed during prolonged cold storage.

### 3.4. Antibiotic Resistance

The resistance patterns to 12 antibiotics in 83 non-psychrotolerant *B. cereus* isolates and 18 psychrotolerant *B. cereus* isolates were determined, and the results are shown in Table 5. No difference in the frequency of antibiotic resistance was observed between non-psychrotolerant *B. cereus* and psychrotolerant *B. cereus* isolates. All isolates revealed 100% resistance to penicillin and third-generation cephalosporins and were sensitive to gentamicin, imipenem, ciprofloxacin, erythromycin, and chloramphenicol. This result is consistent with that of a previous report demonstrating that *B. cereus* is susceptible to imipenem, vancomycin, chloramphenicol, ciprofloxacin, erythromycin, and gentamicin [23,45]. In our study, all non-psychrotolerant *B. cereus* and psychrotolerant *B. cereus* isolates showed resistance to beta-lactam antibiotics. *B. cereus* generally produces beta-lactamase and is uniformly resistant to beta-lactam antibiotics including third-generation cephalosporins. *B. cereus* isolated from ice cream showed sporadic resistance to tetracycline and erythromycin [22].

In this study, 16% of the non-psychrotolerant *B. cereus* isolates showed resistance to tetracycline, and 60% of the isolates revealed an intermediate level of resistance to this antibiotic. These findings are consistent with those of other researchers who reported that 7% of the isolates from cereal, ice cream, and soil were resistant to tetracycline [22]. A high number (60%) of psychrotolerant *B. cereus* isolates showed an intermediate level of resistance to tetracycline. Intermediate level resistance (low-level resistance) is often underestimated. It is not only a gateway to high-level clinical resistance, but is also a gateway to often unsuspected phenomena such as increased bacterial adaptation to antibiotics [46]. Antibiotic resistance in foodborne pathogens has become a serious problem in many parts of the world, emphasizing the need to better understand the mechanisms involved in the emergence and spread of resistant strains [46].

Rifampin is a bactericidal antibiotic effective against a wide variety of bacteria, but *B. cereus* is frequently resistant to rifampin [47]. In this study, 78% of non-psychrotolerant *B. cereus* and 22% of psychrotolerant *B. cereus* isolates were resistant to rifampin. *B. cereus* isolates from agricultural products showed a high frequency of resistance to rifampicin [21], and more than 60% of *B. cereus* isolates from cereal and rice have been shown to be resistant to rifampicin [48]. In contrast, isolates from dairy products and ready-to-eat products show sensitivity to rifampin [49,50]. Only one psychrotolerant *B. cereus* isolate showed resistance to vancomycin. All non-psychrotolerant *B. cereus* isolates were sensitive to clindamycin, whereas 45% of the psychrotolerant *B. cereus* isolates displayed intermediate resistance to clindamycin.

Antibiotic therapy is still the primary treatment method for infections of *B. cereus*. However, antibiotic-resistant *B. cereus* can easily spread at each stage of the food chain, and may reach humans indirectly through the consumption of contaminated food or food derived products [51]. Under suitable conditions, some bacteria can either induce the onset of a disease or transfer the gene that provides antibiotic resistance to other bacterial pathogens [52]. Furthermore, when bacteria are exposed to environmental stress, they may undergo phenotypic and genotypic changes to enhance their survival in a stressful environment [53]. These changes may give rise to cross-protection when these bacteria are exposed to secondary stresses, which may also subsequently change their antibiotic resistance profiles [54]. McMahon et al. found that when *Escherichia coli*, *Salmonella enterica* serovar Typhimurium, and *S. aureus* were exposed to sublethal food preservation stresses, their antibiotic sensitivity profiles changed [55]. The difference in the antibiotic susceptibility profile between non-psychrotolerant *B. cereus* and psychrotolerant *B. cereus* isolates may be due to changes in the phenotype and genotypic properties of -low-temperature-tolerant bacterial isolates.

### 3.5. Biofilm Formation

A biofilm is a multicellular complex formed by microorganisms that are attached to a surface and are embedded in a matrix consisting of exopolymeric substances (EPS). Cells within a biofilm are surrounded by the EPS matrix and cells in the outer layers of the biofilm, protecting them from harsh influences from the environment. Therefore, cells surrounded by biofilms can be resistant to antimicrobial substances or cleaning agents [56]. The ability of non-psychrotolerant *B. cereus* and psychrotolerant *B. cereus* isolates to produce biofilms on a microtiter plate at 7 °C, 10 °C, and 30 °C is shown in Figure 2. At 30 °C, five (6.1%) of the non-psychrotolerant *B. cereus* isolates had moderate biofilm formation abilities, and 19.5% of the isolates had weak biofilm formation abilities. At 10 °C, four isolates (approximately 5%) were moderate biofilm formers, and 25 isolates (30.5%) were weak biofilm formers. Ten percent of non-psychrotolerant *B. cereus* isolates had a weak biofilm-formation ability, but most of the isolates (90%) did not produce a biofilm at 7 °C. No isolate was a strong biofilm former among the entire non-psychrotolerant *B. cereus* isolates.

In contrast, psychrotolerant *B. cereus* isolates showed a higher biofilm formation ability at 7 °C than at 10 °C or 30 °C. Among psychrotolerant *B. cereus* isolates, all were non- or weak biofilm formers at 30 °C, and 38.5% of the isolates had moderate biofilm-forming ability at 10 °C. Interestingly, of the psychrotolerant *B. cereus* isolates, at 7 °C, 23% were strong and moderate biofilm formers, and 15% were weak biofilm formers. Only three psychrotolerant *B. cereus* isolates (16.6%) were strong biofilm formers at 7 °C, in contrast to the psychrotolerant *B. cereus* isolates with no strong biofilm former.

Cells in biofilm are more resistant to disinfectant treatment and stress conditions, and the biofilm formation of some psychrotolerant *B. cereus* isolates under low temperature can be difficult to disinfect foodstuff or food contact surfaces and the possibility of cross-contamination during food processing [57,58]. In this study, psychrotolerant *B. cereus* isolates had greater biofilm-formation abilities than non- psychrotolerant *B. cereus* isolates at low temperatures. This result is in agreement with that of Sarita et al. [57], who showed that *B. cereus* isolates from a chiller tank containing pasteurized milk were able to form a biofilm even at 4 °C. In contrast, *B. cereus* isolated from food products could grow at low temperatures, but the isolates were unable to form a biofilm at 7 °C [28]. The growth of most *B. cereus* isolates is generally inhibited when temperature is below 10 °C and forms strong biofilms on a microtiter plate at 30 °C. In this study, however, some psychrotolerant *B. cereus* isolates could grow at refrigerated temperature and these isolates in biofilm were more resistant to low temperatures than optimal temperature, especially for the transport, retail, and processing of agricultural products such as vegetables, which are typically stored between 0–10 °C, where psychrotolerant *B. cereus* adapted to cold stress and protected themselves against the cold shift. The results suggest that low temperature may fail to restrain the growth and biofilm formation of some *B. cereus* isolates in vegetables, therefore presents a potential risk in food safety.

Biofilm formation is enhanced by cell motility, particularly when biofilm is mediated by flagella, and under certain environmental conditions, flagella are necessary for *B. cereus* biofilm formation [59]. The integration of many diverse signals from the environment, together with other events such as phenotypic and genetic switching during biofilm production and the release of extracellular polymeric substances, might play a role in biofilm formation [60]. Thus, the biofilm-formation ability of psychrotolerant isolates could be due to modified bacterial characteristics (flagella, EPS, growth phase, metabolic activity, etc.) that are temperature dependent. However, further study is necessary to understand the mechanism of biofilm formation by psychrotolerant *B. cereus* isolates.

## 4. Conclusions

*B. cereus* is the predominant species isolated from agricultural products including green leaf lettuce. Low-temperature-tolerant *B. cereus* isolates are associated with potential health risks as these isolates share the enterotoxin gene with mesophilic isolates. Psychrotolerant *B. cereus* isolates can also form strong biofilms, which hampers food industry surface decontamination and increases the possibility of cross-contamination during food processing. The prevalence and pathogenicity of psychrotolerant *B. cereus* during the cold chain cannot be ignored from the aspect of food safety.

## Figures and Tables

**Figure 1 foods-09-00249-f001:**
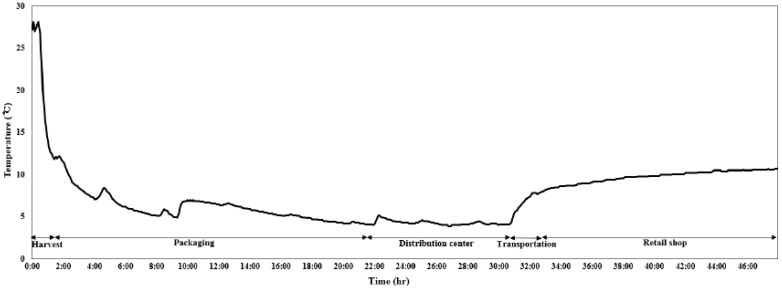
Product temperature recorded by temperature sensors during the cold chain.

**Figure 2 foods-09-00249-f002:**
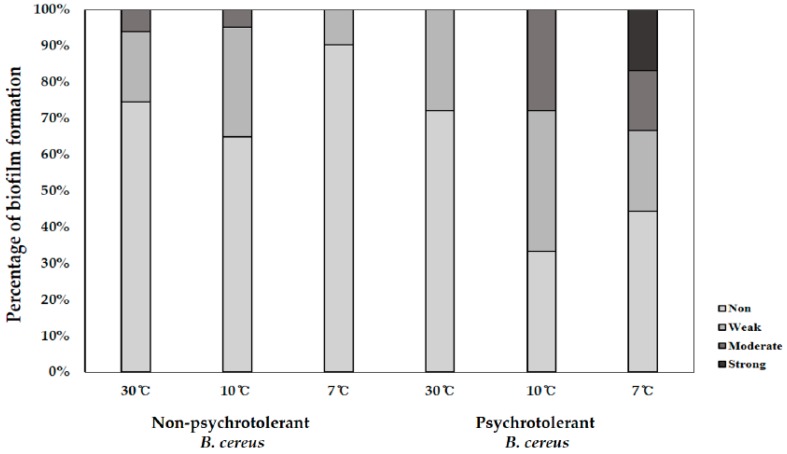
Biofilm phenotype (%) of non-psychrotolerant and psychrotolerant *B. cereus* isolated from green leaf lettuce in the cold chain.

**Table 1 foods-09-00249-t001:** Sequences of the primers used in this study.

Targent Gene	Primer	Sequence (5’–3’)	Melting Temp (°C)	Amplicon (bp)
*hblA*	hblA-F hblA-R	GTG CAG ATG TTG ATG CCG AT ATG CCA CTG CGT GGA CAT AT	55	319
*hblC*	hblC-F hblC-R	AAT GGT CAT CGG AAC TCT AT CTC GCT GTT CTG CTG TTA AT	55	749
*hblD*	hblD-F hblD-R	AAT CAA GAG CTG TCA CGA AT CAC CAA TTG ACC ATG CTA AT	55	429
*nheA*	nheA-F nheA-R	TAC GCT AAG GAG GGG CA GTT TTT ATT GCT TCA TCG GCT	55	499
*nheB*	nheB-F nheB-R	CTA TCA GCA CTT ATG GCA G ACT CCT AGC CGG TGT TCC	55	769
*nheC*	nheC-F nheC-R	CGG TAG TGA TTG CTG GG CAG CAT TCG TAC TTG CCA A	55	581
*entFM*	entFM-F entFM-R	ATG AAA AAA GTA ATT TGC AGG TTA GTA TGC TTT TGT GTA ACC	55	1269
*cytK*	cytK-F cytK-R	GTA ACT TTC ATT GAT GAT CC GAA TAC TAA ATA ATT GGT TTC C	44	505
*ces*	ces-F ces-R	GGT GAC ACA TTA TCA TAT AAG GTG GTA AGC GAA CCT GTC TGT AAC AAC A	55	1271

**Table 2 foods-09-00249-t002:** Contamination level and identification of the *B. cereus* group by the polymerase chain reaction (PCR) method and microbial counts of the *B. cereus* group at each point during the cold chain.

Sampling Point	No. of *B. cereus* Group Isolates (%)			Contamination Level (Log CFU/g)
	*B. cereus*	*B. thuringiensis*	*B. mycoides*	
Harvest	29 (100)	0 (0)	0 (0)	1.9 ± 0.22 ^a,(1),(2)^
Packaging	17 (100)	0 (0)	0 (0)	2.9 ± 0.44 ^b^
Distribution center	15 (100)	0 (0)	0 (0)	2.6 ± 0.68 ^b^
Transportation	10 (100)	0 (0)	0 (0)	2.1 ± 0.26 ^ab^
Retail shop	30 (100)	0 (0)	0 (0)	2.6 ± 0.41 ^b^
Total	101 (100)	0 (0)	0 (0)	2.4 ± 0.39

^(1)^ Each data point shows the mean ± standard deviation of triplicated samples. ^(2)^ Different superscripts in the same column indicate significant differences (*p* < 0.05).

**Table 3 foods-09-00249-t003:** Evaluation of the growth of *Bacillus cereus* isolated from green leaf lettuce incubated at different temperatures.

Sampling Point	No. of *B. cereus* Isolates	Temperature of Growth (°C)			
		12 °C	10 °C	7 °C	5 °C
Harvest	29	10 (34.5)	8 (27.6)	0 (0)	0 (0)
Packaging	17	12 (70.6)	12 (70.6)	1 (5.9)	0 (0)
Distribution center	15	14 (93.3)	14 (93.3)	6 (33.3)	0 (0)
Transportation	10	7 (70)	5 (50.0)	4 (20.0)	0 (0)
Retail shop	30	16 (53.3)	12 (40.0)	7 (23.3)	0 (0)
Total	101	59 (58.4)	51 (50.5)	18 (17.8)	0 (0)

**Table 4 foods-09-00249-t004:** Frequency and profile of enterotoxigenic non-psychrotolerant and psychrotolerant *B. cereus* isolated from green leaf lettuce in the cold chain.

	Enterotoxin Genes	No. (%) of Entertoxigenic *B. cereus*	
		Non-psychrotolerant *B. cereus* (*n* = 83)	Psychrotolerant *B. cereus* (*n* = 18)
Frequency of Enterotoxin Genes			
1	*nheABC*	81 (97.6)	17 (94.4)
2	*hblACD*	25 (30.1)	8 (44.4)
3	*cytK*	80 (96.4)	10 (55.6)
4	*entFM*	73 (88.0)	14 (77.8)
5	*ces*	0 (0.0)	0 (0.0)
Profile of Enterotoxin Genes			
1	*nheABC + hblACD + cytK + entFM*	19 (23.2)	5 (27.8)
2	*nheABC + hblACD + entFM*	0 (0.0)	3 (16.7)
3	*nheABC + cytK + entFM*	52 (63.4)	3 (16.7)
4	*nheABC + hblACD + entFM*	2 (2.4)	0 (0.0)
5	*nheABC + hblACD + cytK*	3 (3.7)	0 (0.0)
6	*nheABC + cytK*	4 (4.9)	1 (5.6)
7	*nheABC + entFM*	0 (0.0)	1 (5.6)
8	*cytK + entFM*	2 (2.4)	1 (5.6)
9	*entFM*	0 (0.0)	1 (5.6)
10	*nheABC*	1 (1.2)	3 (16.7)

**Table 5 foods-09-00249-t005:** Antibiotic resistance patterns of the non-psychrotolerant and psychrotolerant *B. cereus* isolated from green leaf lettuce in the cold chain.

	No. (%) of *B. cereus* Isolates with Indicated Response					
	Non-psychrotolerant *B. cereus* (*n* = 83)			Psychrotolerant *B. cereus* (*n* = 18)		
	Resistance	Intermediate	Sensitive	Resistance	Intermediate	Sensitive
Tetracycline	13 (15.7)	50 (60.2)	20 (24.1)	2 (11.1)	3 (16.7)	13 (72.2)
Gentamicin	0 (0.0)	0 (0.0)	83 (100.0)	0 (0.0)	0 (0.0)	18 (100.0)
Imipenem	0 (0.0)	0 (0.0)	83 (100.0)	0 (0.0)	0 (0.0)	18 (100.0)
Vancomycin	1 (1.2)	0 (0.0)	82 (98.8)	0 (0.0)	0 (0.0)	18 (100.0)
Ciprofloxacin	0 (0.0)	0 (0.0)	83 (100.0)	0 (0.0)	0 (0.0)	18 (100.0)
Erythromycin	0 (0.0)	0 (0.0)	83 (100.0)	0 (0.0)	0 (0.0)	18 (100.0)
Rifampin	65 (78.3)	13 (15.7)	5 (6.0)	4 (22.2)	6 (33.3)	8 (44.4)
Chloraphenicol	0 (0.0)	0 (0.0)	83 (100.0)	0 (0.0)	0 (0.0)	18 (100.0)
Clindamycin	0 (0.0)	0 (0.0)	83 (100.0)	0 (0.0)	8 (44.4)	10 (55.6)
Penicillin	83 (100.0)	0 (0.0)	0 (0.0)	18 (100.0)	0 (0.0)	0 (0.0)
Ceftriaxone	83 (100.0)	0 (0.0)	0 (0.0)	18 (100.0)	0 (0.0)	0 (0.0)
Cefotaxim	83 (100.0)	0 (0.0)	0 (0.0)	18 (100.0)	0 (0.0)	0 (0.0)

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
