# Peer review of "Enterotoxin Genes, Antibiotic Susceptibility, and Biofilm Formation of Low-Temperature-Tolerant Bacillus cereus Isolated from Green Leaf Lettuce in the Cold Chain"

_foods, 2020, doi:10.3390/foods9030249_

Round 1

Reviewer 1 Report

This manuscript presents very interesting conclusions, that are of importance when assessing the risks posed by Bacillus in psycrothrophic food environments. It is generally well written. However, it has one large flaw, that must be corrected throughout the whole text before its publication should be considered: the authors repeatedly confuse "growth" with "survival". A psychrophilic microorganism has the ability to grow under refrigeration conditions; in fact, refrigeration is not meant to be bactericidal, that is, to threaten survival. Furthermore, the Introduction lacks background on two important aspects - one of which is even part of the title of the manuscript - the effect of cold storage on toxin production by the Bacillus species under study and on the relevance of antibiotic resistance in the context of the cold chain (i.e., the possibility of horizontal gene transfer to other bacteria). These must be added.

Regarding antibiotic testing, the best way to assess the risk posed would have been to detect the presence of relevant antibiotic resistance genes in the bacterial isolates under study. This should be added to the manuscript. Simply testing for resistant phenotypes does not provide enough information for risk assessment.

Other minor comments that must also be addressed follow below.

Lines 2, 33 and 49 – The authors should not employ the term “low-temperature surviving” in this context. All Bacillus species can survive low temperatures, meaning that _they retain their viability_ after being submitted to temperatures that are below their minimum for growth, i.e., when conditions are favourable again, they will resume their growth, which was halted during the low temperature exposure. In the bacterial world, refrigeration is regarded as merely bacteriostatic, not bactericidal. What separates certain bacilli from the others regarding their temperature preferences, is the ability (or lack thereof) to _grow_ (meaning: to increase in number) at low temperatures. There is, in the whole text, a confusion between growth and survival that must be corrected here and throughout the whole text. A new acronym must be devised too. LTSBc is not correct, since it is linked with the concept of survival, which does not apply here.

Lines 29 – 73 – The introduction lacks a brief discussion on the effect of low temperatures on toxin production by bacilli. Also, it does not provide a background for the antibiotic resistance testing that appears in the Material and Methods section for the first time. Text containing information on both of these should be added to the Introduction.

Lines 90 and 91 – The authors stated that “The temperature of the vegetables was maintained at 10°C throughout the entire distribution chain”, but the data in Fig. 1 do not support this statement. The text should be rewritten to be in accordance with the data in the aforementioned figure. Also, clarify what is the target (ideal) refrigeration temperature for lettuce – from a general perspective, 10°C seems too high.

Lines 139 – 148 – Antibiotic resistance testing is relevant in this context, but the Introduction does not provide the objective(s) for including these tests in the manuscript. A short text should be inserted there to cover this gap. Also, it would be important to test for the presence/absence of relevant antibiotic resistance genes in the isolates under study, as these can, specially if present in mobile genetic elements, be passed to other constituents of the vegetable microbiota during storage, with negative consequences in terms of public health.

Table 2 – Change B. mycoid to B. mycoides

Lines 194 – to 197 – The way this sentence is formulated “Although the samples examined in this study were raw…” may give the impression that, if they were to be subsequently cooked, the hazard might be eliminated, which is not true. First, cooking by the usual procedures would not eliminate the hazards linked to the presence of sporeforming bacteria such as Bacillus. Second, lettuce is mostly consumed raw. The authors should change the text to avoid these possible interpretations.

Lines 209 – 2019 – There is again the same confusion between “survival” and “growth” at low temperatures. The authors should correct the text and should replace, in all instances, what they call “survival” with “growth”.

Lines 222 – 223 – The comparison with milk is out of place here. The authors should rather refer to studies in vegetables, if available.

Line 223 – The word psychrotrophic is misspelled. Please correct it.

Lines 256 – 293 – A discussion on the potential for the horizontal transfer of antibiotic resistance genes in the bacilli is lacking. Preferably, this should be substantiated with further data on the presence/absence of such genes in the genome of the isolates under study.

Author Response

Reviewer 1

This manuscript presents very interesting conclusions, that are of importance when assessing the risks posed by Bacillus in psycrothrophic food environments. It is generally well written. However, it has one large flaw, that must be corrected throughout the whole text before its publication should be considered: the authors repeatedly confuse "growth" with "survival". A psychrophilic microorganism has the ability to grow under refrigeration conditions; in fact, refrigeration is not meant to be bactericidal, that is, to threaten survival.

Thank you for your comments. I agree with your opinion. I understood the characteristics of mesophilic B. cereus and psychrotolerant B. cereus and revised the contents to ‘growth’ from ‘survival’ in overall manuscript.

Furthermore, the Introduction lacks background on two important aspects - one of which is even part of the title of the manuscript - the effect of cold storage on toxin production by the Bacillus species under study and on the relevance of antibiotic resistance in the context of the cold chain (i.e., the possibility of horizontal gene transfer to other bacteria). These must be added.

I discussed about the toxin production pattern of psychrotolerant B. cereus under cold stress and the importance of antibiotic resistant B. cereus in food industry. These contents added in Introduction, Results and Discussion.

 Regarding antibiotic testing, the best way to assess the risk posed would have been to detect the presence of relevant antibiotic resistance genes in the bacterial isolates under study. This should be added to the manuscript. Simply testing for resistant phenotypes does not provide enough information for risk assessment.

Thank you for your comments. The antibiotic resistance pattern was achieved according to the protocols of Clinical and Laboratory Standard Institute guideline. Agar disk diffusion testing approved by CLSI is the official method used in microbiology laboratories for routine antibiotic susceptibility testing. In present study, we did not consider the horizontal gene transfer to other bacteria. Because antibiotic resistance of B. cereus isolates underreported yet, thus we thought that it is important to evaluate the resistance pattern to antibiotics of B. cereus isolates from various foodstuff to provide comprehensive information about antibiotic resistance pattern of B. cereus, especially psychrotolerant B. cereus. However, our further study will consider the presence and transfer of antibiotic resistant gene in order to evaluate the genotypic resistance pattern.

Other minor comments that must also be addressed follow below.

Lines 2, 33 and 49 – The authors should not employ the term “low-temperature surviving” in this context. All Bacillus species can survive low temperatures, meaning that _they retain their viability_ after being submitted to temperatures that are below their minimum for growth, i.e., when conditions are favourable again, they will resume their growth, which was halted during the low temperature exposure. In the bacterial world, refrigeration is regarded as merely bacteriostatic, not bactericidal. What separates certain bacilli from the others regarding their temperature preferences, is the ability (or lack thereof) to _grow_ (meaning: to increase in number) at low temperatures. There is, in the whole text, a confusion between growth and survival that must be corrected here and throughout the whole text. A new acronym must be devised too. LTSBc is not correct, since it is linked with the concept of survival, which does not apply here.

I agree with your opinion. I revised the title to “low-temperature-tolerant (psychrotolerant)” from “low-temperature-surviving” in overall manuscript.

Lines 29 – 73 – The introduction lacks a brief discussion on the effect of low temperatures on toxin production by bacilli. Also, it does not provide a background for the antibiotic resistance testing that appears in the Material and Methods section for the first time. Text containing information on both of these should be added to the Introduction.

I inserted the contents about antibiotic resistance of B. cereus strains in Introduction.

Lines 90 and 91 – The authors stated that “The temperature of the vegetables was maintained at 10°C throughout the entire distribution chain”, but the data in Fig. 1 do not support this statement. The text should be rewritten to be in accordance with the data in the aforementioned figure. Also, clarify what is the target (ideal) refrigeration temperature for lettuce – from a general perspective, 10°C seems too high.

I revised and rewritten the result about temperature of vegetables during cold chain in Fig. 1.

Lines 139 – 148 – Antibiotic resistance testing is relevant in this context, but the Introduction does not provide the objective(s) for including these tests in the manuscript. A short text should be inserted there to cover this gap. Also, it would be important to test for the presence/absence of relevant antibiotic resistance genes in the isolates under study, as these can, specially if present in mobile genetic elements, be passed to other constituents of the vegetable microbiota during storage, with negative consequences in terms of public health.

As mentioned above, since antibiotic resistance of B. cereus isolates underreported yet, thus I thought that it is important to evaluate the resistance pattern to antibiotics of B. cereus isolates from various foodstuff to provide comprehensive information about antibiotic resistance pattern of B. cereus, especially psychrotolerant B. cereus. However, our further study will consider the presence and transfer of antibiotic resistant gene in order to evaluate the genotypic resistance pattern. I discussed and added the importance of evaluation of antibiotic resistance in Introduction.

Table 2 – Change B. mycoid to B. mycoides

I corrected.

Lines 194 – to 197 – The way this sentence is formulated “Although the samples examined in this study were raw…” may give the impression that, if they were to be subsequently cooked, the hazard might be eliminated, which is not true. First, cooking by the usual procedures would not eliminate the hazards linked to the presence of sporeforming bacteria such as Bacillus. Second, lettuce is mostly consumed raw. The authors should change the text to avoid these possible interpretations.

I rewritten the sentence without possible interpretation.

Lines 209 – 2019 – There is again the same confusion between “survival” and “growth” at low temperatures. The authors should correct the text and should replace, in all instances, what they call “survival” with “growth”.

I replace to “growth” or “grow” from “survival”.

Lines 222 – 223 – The comparison with milk is out of place here. The authors should rather refer to studies in vegetables, if available.

I delete the contents about the comparison with milk. I added the results from other researchers in Results and Discussion.

Line 223 – The word psychrotrophic is misspelled. Please correct it.

I corrected.

Lines 256 – 293 – A discussion on the potential for the horizontal transfer of antibiotic resistance genes in the bacilli is lacking. Preferably, this should be substantiated with further data on the presence/absence of such genes in the genome of the isolates under study.

Detailed investigations of antibiotic resistant B. cereus in various foodstuff remain scarce. It is therefore necessary to acquire a better and more accurate understanding of the antibiotic resistant pattern in B. cereus isolates from foodstuff, especially, environmental stress tolerant isolates. One of the objective in this study was to define the diversity and variation of the B. cereus isolates from representative green leafy vegetable and improve the understanding of the differences between non-psychrotolerant and psychrotolerant B. cereus.  

Reviewer 2 Report

Title: Enterotoxin Genes, Antibiotic Susceptibility, and Biofilm Formation of Low-Temperature-Surviving Bacillus cereus Isolated from Green Leaf Lettuce in the Cold Chain

The authors studied the prevalence and characteristics of LTSBc strains associated with green leafy vegetables. Generally the manuscript is well structured and written. My major concerns are: the risk presented by foodborne B. cereus is very low. This should be stated and appropriate references provided (e.g., Webb et al 2019). Please justify your focus; provide recent reported illness outbreaks due to the consumption of fresh or minimally processed foods specifically lettuce contaminated with this microorganism. How statistical analysis was performed is not clear as the whole section in the methodology is absent.

Minor comments  

Line 35: substitute ‘in’ with ‘during’

Line 82: indicate storage duration at 9-10 °C

Line 123 – 124: ‘Bacteria with rhizoidal growth were classified as B. mycoides without further discrimination’ please specify the reason

Line 128: ‘TSA plate and was incubated at 7 °C for 20 days and then at 10 °C for 12 days’ please clarify this sequence of temperature regimes. State the incubation period for temperature condition of 5 °C.

Line 139 - 148: antibiotic susceptibility testing how the results are presented?

Line 168: Clarify if an empty well was used to read the absorbance

Line 160 – 167 and 168 – 174: provide appropriate references for biofilm evaluation procedure

Line 202: ‘physiological attributes’ not clear what author meant to say. Clarify is this about food or bacteria. Expand by providing examples.

Line 210: 58, 50, 18% - where this number is in table 3

Table 3 – indicate in the table heading the meaning of the number and number in brackets

Table 5 title  - change appropriately indicating what your results mean

Author Response

Reviewer 2

The risk presented by foodborne B. cereus is very low. This should be stated and appropriate references provided (e.g., Webb et al 2019). Please justify your focus; provide recent reported illness outbreaks due to the consumption of fresh or minimally processed foods specifically lettuce contaminated with this microorganism. How statistical analysis was performed is not clear as the whole section in the methodology is absent.

 Thank you for your comments. The incidence of B. cereus in food has been widely reported because B. cereus is ubiquitously found in the environment and thus easily contaminate food. B. cereus is generally present in the soil and can cause food poisoning in fresh vegetables. Outbreaks with vomiting and diarrheal syndromes caused by B. cereus have been reported in Finland (Shaheen et al. 2010), Belgium (Andreja et al., 2010), Thailand (Chitov et al., 2008), the United Kingdom (Meldrum et al. 2009; Altayar and Sutherland, 2010), the United States (Ankolekar et al., 2009), South Korea (Park et al., 2009), and Africa (Ouoa et al.2008), France and China (Glasset et al., 2016; Paudyal et al., 2018). Especially, food poisoning outbreaks associated with vegetables contaminated by foodborne pathogens in Korea increased from 119 in 1998 to 271 in 2010 and the most frequent foodborne pathogen found in vegetables was B. cereus (Park et al. 2018). Therefore, it is necessary to monitor the contamination level of B. cereus in vegetables. These contents inserted in Introduction. Statistical analysis against microbiological test also inserted in the Materials and Methods.

Minor comments  

Line 35: substitute ‘in’ with ‘during’

I revised to “during”

Line 82: indicate storage duration at 9-10 °C

I revised the sentence according to your comments

Line 123 – 124: ‘Bacteria with rhizoidal growth were classified as B. mycoides without further discrimination’ please specify the reason

Bacillus mycoides has a distinct characteristics of filamentous and rhizoid colonies formed on agar plate (ÅšwiÄ™cicka, I et al. 2006). We identified B. mycoides isolates by morphology confirmation.  

Line 128: ‘TSA plate and was incubated at 7 °C for 20 days and then at 10 °C for 12 days’ please clarify this sequence of temperature regimes. State the incubation period for temperature condition of 5 °C.

I revised the method about temperature treatment.

Line 139 - 148: antibiotic susceptibility testing how the results are presented?

The results were interpreted as ‘susceptible’, ‘intermediate’, and ‘resistant’ in accordance with the criteria provided by the CLSI (Performance standards for antimicrobial susceptibility testing; Twenty-Fourth Informational Supplement; CLSI document M100-S24).

Line 168: Clarify if an empty well was used to read the absorbance

I performed the biofilm evaluation of B. cereus ATCC 14579 and TSB broth as negative control.

Line 160 – 167 and 168 – 174: provide appropriate references for biofilm evaluation procedure

I inserted the reference about the method of biofilm evaluation procedure.

Line 202: ‘physiological attributes’ not clear what author meant to say. Clarify is this about food or bacteria. Expand by providing examples.

I inserted the detail contents about physiological attributes under cold stress.

Line 210: 58, 50, 18% - where this number is in table 3

I revised the number to 58.4% at 12°C, 50.5% at 10°C and 17.8% at 7°C from 58%, 50% and 18%, respectively in Result and Discussion.

Table 3 – indicate in the table heading the meaning of the number and number in brackets

I deleted the table heading.

Table 5 title  - change appropriately indicating what your results mean

I revised the title of Table 5 to “Antibiotic resistance patterns of non- non-psychrotolerant and psychrotolerant B. cereus isolated from green leaf lettuce in the cold chain”.
